# Quality of Life in Healthcare Workers during COVID-19—A Longitudinal Study

**DOI:** 10.3390/ijerph20146397

**Published:** 2023-07-19

**Authors:** Robin Jungmar Ridell, Lotti Orvelius

**Affiliations:** 1Department of Infectious Diseases in Östergötland and Department of Biomedical and Clinical Sciences, Linköping University, 581 83 Linköping, Sweden; 2Department of Intensive Care, Clinical and Experimental Medicine, Linköping University, 581 83 Linköping, Sweden; lotti.orvelius@regionostergotland.se

**Keywords:** RAND-36, physicians, registered nurses, licensed practical nurses, daily life, pandemic, work environment

## Abstract

The COVID-19 pandemic occurred in 2020, and affected people’s daily life worldwide at work and at home. Healthcare workers are a professional group with heavy workloads, and during the COVID-19 pandemic, their burden increased. The literature from earlier outbreaks describes risks for affected mental health in frontline workers, and the main aim of this study is to examine healthcare workers’ quality of life during the COVID-19 pandemic. In addition, we sought to assess if there was any difference in working at a pandemic ward compared to anon-pandemic ward. In this longitudinal and descriptive study, a total of 147 healthcare workers assessed their perceived health every third month over one year using the RAND-36 health survey. RAND-36 is a general instrument that consists of 36 questions and is widely used for assessing quality of life. The healthcare workers in this study showed reductions in perceived quality of life during the first six months of the COVID-19 pandemic. Healthcare workers on a pandemic ward reported a lower score in RAND-36 compared to healthcare workers on a non-pandemic ward. Registered nurses and licensed practical nurses seemed more negatively affected in their quality of life than physicians. Compared to data from the general Swedish population, healthcare workers in this study had less energy during this period.

## 1. Introduction

### 1.1. Literature Background

At the start of 2020, the disease known as COVID-19 was spread around the world, resulting in a worldwide pandemic. Many people affected by COVID-19 needed hospital care and the burden on public health systems increased. The availability of healthcare workers, hospital beds, intensive care units, and respirator resources were vital to be able to treat these patients, and the need was expected to increase in correlation with the increased number of COVID-19 cases, exceeding capacity [1]. To date, there have been over 757 million confirmed cases of COVID-19 worldwide, and more then 6.9 million confirmed deaths, according to the world health organization [2].

The workload and distress increased for healthcare workers during the COVID-19 pandemic, and the burden on healthcare workers can have a serious impact on healthcare workers’ mental health and quality of life. Previous studies have shown that the ongoing stress that they faced may have had negative effects on their psychological well-being, and may have affected the quality of care for patients and the practices of healthcare workers [2,3]. In addition, the heavily burdened health system may have led to increased risks to patients’ safety [4]. 

High workload, shift work, sleep disorders, job satisfaction and years in a profession often affect the risk of mental illness for staff [5]. Continuous and increased stress can quickly accelerate when healthcare workers feel that there is not enough time or limited resources to care for patients properly. Extreme stress may lead to insomnia, fatigue, irritation, anxiety, and depression [6], and nurses regularly experience a variety of work-related stressors such as long shifts, irregular schedules, lack of professional support and the added work necessary to meet the patients’ needs. Depressive symptoms among nurses have been reported to be between 18% and 41%, respectively, in two studies [2,3], and for physicians approximately 28% [7]. 

Furthermore, epidemic outbreaks may cause trauma for healthcare workers and affect mental health, as the large number of patients, the lack of personal protective equipment and the fear of a new disease increase. Fear, anxiety, and helplessness cause physical exhaustion similar to what is found in studies on the outbreak of Ebola and MERS-CoV. The literature describes that positive and negative feelings occurred in frontline nurses working with COVID-19 patients in China. In the beginning of the outbreak, the negative feelings dominated, while the positive feelings developed over time. During stress, healthcare workers felt that support and help for healthcare workers resulted in stress relief, and that the collective power was a positive feeling [8]. Nurses perceive that the quality of care and patients’ safety is reduced when there are a lot of hours spent working overtime and a heavy workload [9]. 

Quality of life in healthcare workers may be affected by conflicts with colleagues, patient-related errors, unmet expectations from patient’s caregivers and dealing with human suffering. Good coping strategies are associated with a better quality of life. Job satisfaction correlates with quality of life and fatigue affects quality of life negatively [10].

### 1.2. Study Aim

Many studies have shown that healthcare workers’ quality of life (QoL) and mental health are affected when the burden raises, and therefore the main aim of this study is to examine healthcare workers’ quality of life with RAND-36 during one year of the COVID-19 pandemic at a hospital in Sweden. A secondary aim was to investigate whether there was any difference between workers at pandemic ward and workers at a non-pandemic ward.

## 2. Material and Methods

### 2.1. Design

This study was conducted with a longitudinal and descriptive design; the study samples consisted of data on healthcare workers in five different wards at a larger university hospital in the southeast of Sweden. The hospitals have approximately 300 hospital beds and almost 5500 employees with a health service area for approximately two million people. Physicians, registered nurses, and licensed practical nurses who cared for patients with a COVID-19 diagnosis on a pandemic ward were compared with physicians, registered nurses, and licensed practical nurses on non-main pandemic wards. All included participants were asked to fill out the RAND-36 at the start of the study, and again every third month for a year. Demographic and work-related data were obtained at the first measurement point. The data included: sex, age, profession, years in profession, type of ward, work experience in the current ward, employment status and “sick leave in the last 12 months?” (yes/no). Healthcare workers who had taken sick leave in the last twelve months at the start of the study were excluded. 

### 2.2. Instrument

The RAND-36 item health survey is used for assessing quality of life [11]. RAND-36 has been translated into and validated for the Swedish language [12], and consists of 36 questions in which 35 questions are split into one of two domains: mental and physical health. These domains consist of eight health dimensions. The dimensions are physical functioning (PF), role limitations caused by physical health problems (RP), role limitations caused by emotional problems (RE), social functioning (SF), emotional well-being (MH), energy/fatigue (VT), bodily pain (BP), and general health (GH) perceptions. Each dimension scores on a range of 0 to 100, in which a higher score represents higher perceived health status.

### 2.3. Procedure

The data were collected during the pandemic in Sweden starting in late June 2020 and follow-up was measured every third month until the end of the study, in May 2021. Information about the study was given on the wards and the health survey was administered there in a paper format. The participants of the study wrote consent to participate and were allowed to terminate the study at any time. Each participant had a coded number instead of their name being used. The study was anonymous and confidential. The follow-up measurements were requested by e-mail, anonymously through an internet-based survey program, with weekly reminders for up to one month. Early on, two wards, the infection ward and the intensive care unit, were determined to be pandemic wards at the hospital. These were included as pandemic wards, and the cardiology department, thoracic clinic and orthopedic ward were included as non-pandemic wards. These wards were ascertained to not care for COVID-19 patients in the beginning, but during the pandemic half the orthopedic ward converted to a pandemic ward, and therefore was analyzed as a mixed ward instead. It was thus excluded from the comparison of pandemic and non-pandemic wards.

### 2.4. Data Analysis

Statistical analysis of the study was performed using SPSS version 27 (IBM Corporation, New York, NY, USA). Descriptive data are presented as percent, mean and standard deviation, and range. To analyses different measurement occasions and individuals, the paired sample *t* test was used. One-way analysis of variance was used for comparison between demographic data and different dimensions of RAND-36, and a general linear model was used for the comparison of different occasions and groups. All statistical results were evaluated at 95% confidence intervals and a significance level of *p* < 0.05 was accepted. For the comparison of RAND-36 with a general population, normal data from a previous publicized study were used (12). Data included 3422 participants, 55% of which were females and 45% were male, which is analogous to the Swedish general population. The age of the participants did differ slightly from the Swedish population, with a greater representation being 60 years and older. The study was conducted in a longitudinal and descriptive design without a testing hypothesis, and therefore no Bonferroni correction was needed.

## 3. Results

### 3.1. Demographics

At the first measurement (baseline, month 0), 147 out of 500 (29.4%) healthcare workers responded. A total of 81.5% were women (n = 120), the mean age of the participants was 38.2 +/− 13 and the mean amount of years spent working was 13 +/− 12 years. A total of 15.1% (n = 22) worked as physicians, 44.5% (n = 64) worked as registered nurses and 40.4% (n = 61) worked as licensed practical nurses (Table 1). Of the participants, 28.2% (n = 42) worked in the cardiac department, 21.2% (n = 31) in the infection ward, 20.5% (n = 30) in the intensive care unit, 15.7% (n = 23) in the thoracic clinic and 14.5% (n = 21) in the orthopedic clinic (Table 1) At the second time of measurement (month 4), 73(49.7%) individuals responded; at the third time of measurement (month 8) 51 (34.7%) responded; and at the fourth and last time of measurement (month 12), 39 (26.5%) individuals responded (Figure 1).

### 3.2. Comparison of Perceived Level of QoL Measurements One and Two

There was a significant impairment between month 0 (baseline) and month 4 for those who answered on both occasions (n = 73); the QoL dimensions general health (GH), energy/fatigue (VT) and mental health (MH) were significant affected negatively. General health (GH) mean scores decreased from 73.6 to 67.4 (*p* = 0.003), energy/fatigue (VT) decreased from 54.4 to 48.2 (*p* = 0.006), and mental health (MH) decreased from 73.3 to 67.9 (*p* = 0.0014) between month 0 (baseline) and month 4, respectively. There was no statistical difference between the other QoL dimensions between month 0 (baseline) and month 4 (Table 2). 

For the single wards/clinics, there was a significant impairment in energy/fatigue (*p* = 0.045), with the mean of 48.92 decreased to a mean of 38.21 for workers on the infection ward, and for the intensive care unit there was a significant impairment in the dimension general health(*p* = 0.017), with the mean of 76.31 decreased to a mean of 67.63. No other significant changes for the orthopedic department, thoracic clinic, or cardiology department between month 0 (baseline) and month 4 were seen.

In the comparison between the healthcare workers on the pandemic wards (infection ward and the intensive care unit) and non-pandemic wards (thoracic clinic and department of cardiology), a significant difference was seen in energy/fatigue, with a mean of 46.66–38.78 compared with 56.25–58.33 (*p* = 0.001). The healthcare workers on the non-pandemic wards reported higher levels of energy/fatigue on QoL (Table 3).

Between professions, there was a significant difference in physical functioning (*p* = 0.046), bodily pain (*p* = 0.013), social functioning (*p* = 0.002) and role functioning/emotional (*p* = 0.016). The licensed practical nurses reported the lowest perceived levels in the different dimensions, and the physicians perceived the highest levels of the dimensions (Table 4*).*

In a comparison of measurements one, two, three and four, there was only a significant impairment in the QoL for the responders between measurements one and two, except for the dimension vitality. Individual follow-up showed that significant deterioration occurred in the dimension energy/fatigue mean, which dropped from 55.49 to 47.54 (*p* = 0.032) for responders (n = 51) between measurements one and three. In comparison with the general Swedish population before the pandemic [13], the healthcare workers perceived higher levels in the dimensions physical functioning, bodily pain and general health, but perceived lower levels in the dimensions role functioning/physical, energy/fatigue, social functioning, role functioning/emotional and emotional well-being.

## 4. Discussion

The main aim of this study was to explore healthcare workers’ quality of life (QoL) during the COVID-19 pandemic at a hospital in the southeast of Sweden. There was a significant impairment in self-perceived QoL between measurement one (month 0, baseline) and measurement two (month 4). The first measurement occurred at the end of the first wave of the pandemic, and the pandemic probably affected the psychological dimensions. Several studies explored the mental health of healthcare workers during the COVID-19 pandemic, and these studies suggest that the burnout level and stress was high in front-line nurses and that there was also a mild/moderate level of depression [14]. Healthcare workers during pandemics have reported mild anxiety, mild depressive symptoms, post-traumatic stress disorder [15] and other psychological factors, such as the fear of being infected as a result of close care of patients, fear of infecting family, and the difficulty of watching patients suffer and die [16].

In our study, there was a significantly negative impairment between measurements one and two in the dimension general health and vitality and the domain mental health. This was interpreted to mean that healthcare workers were more affected in their general health and were more exhausted when the second measurement occurred. However, while the measurements continued during the pandemic on every third month, the responders did not report worse outcomes in RAND-36 after the second measurement. The most significant change over time occurred between the first and second measurements. This could be because the healthcare workers found ways to cope with the situation during the pandemic, and felt acclimatized to a new natural state. Other studies have shown that protective factors to reduce risk for decreased mental health include a supportive work organization, knowledge of the disease and adequate personal protective equipment [8,17,18]. During measurements three (month 8) and four (month 12), the personal protective equipment and the disease itself were better provided for, and this may have impacted the outcome. 

Findings regarding the differences between wards show that healthcare workers on the infection ward and the intensive care unit had significant impairments, while the healthcare workers on the non-pandemic wards, in the thoracic and cardiology departments, had no significant impairments. The affected wards were the pandemic wards. These findings show that front-line healthcare workers caring mainly for COVID-19 patients reported a lower score in RAND-36 than healthcare workers caring for other patients. These results have been shown in other studies in which higher levels of psychological distress have been associated with caring for and contact with COVID-19 patients, compared with caring for and contact with non-COVID-19 patients [19,20,21,22]. One study identified that healthcare workers in the emergency department, respiratory department, intensive care unit and infectious department had twice the risk of depression of non-clinical staff [21].

Furthermore, the prevalence of depression and burnout is common among physicians and registered nurses [2,3,23], and medical errors are more common when physicians are depressed compared to non-depressed physicians. Adverse patient events can occur when overtime, sleep deprivation and workload of healthcare workers increases [23]. There was a significant difference in the group professions; these findings are weak because of the amount of responders, but will be interesting to investigate further. Physicians in this study reported at baseline (month 0) an overall higher self-perceived health than the general Swedish population before the pandemic [13]; registered nurses and licensed practical nurses reported at baseline a lower self-perceived health than physicians. Research conducted during earlier outbreaks shows that emergency nurses experiencemore stress than emergency doctors [24].

Furthermore, in this study, a majority of the healthcare workers assigned lower measurements to role functioning/physical, energy/fatigue, social functioning and role functioning/emotional, while they had higher levels of physical functioning, bodily pain and general health compared to the reference data from the Swedish general population, represented in RAND-36 outcomes [13], at the beginning of the study. These self-perceived health issues could indicate signs and risks of burnout, and that the healthcare workers did not have the energy for a social life after work. The social isolation and risk of loneliness could also be related to the fear of transferring the disease to family members, friends and loved ones [14]. Healthcare workers from this study estimated that they had less energy and higher fatigue than the reference data of the Swedish general population, which may have affected their QoL in terms of physical and emotional role functioning and social functioning. However, reference data from the Swedish general population were collected before the worldwide pandemic, and so the pandemic may have impacted the general population’s perceived health since then.

The findings from this study show that healthcare workers’ QoL was affected during the pandemic. It is important to examine the health of healthcare workers, as this can reveal problems that exist within a workplace. If the problems are consolidated and highlighted, then something can be done about them. Findings from previous papers based on other outbreaks of diseases have recommended creating a good, communicative organization with honesty and empowerment, which has problem-solving strategies and tools to enable communication about patient safety concerns to managers in order to establish psychological safety [25,26].

## 5. Conclusions

The quality of life of healthcare workers was negatively affected during the COVID-19 pandemic. Healthcare workers caring for COVID-19 patients reported a lower quality of life compared to healthcare workers on a non-pandemic ward, but this deterioration stopped after six months. Further research in this area could focus on measures and the long-term follow-up of the healthcare workers who were involved in care during the pandemic to identify the need for help.

There are several limitations to this study. First, the sample size was small with a low response frequency, which could be due to the already high pressure and fatigue experienced by healthcare workers during the COVID-19 pandemic. Stress, work, environment and workload could have all resulted in this lack of responders.

The strengths of this study include the fact that the same responders self-reported outcomes over one year, and that the data were collected during different times in the pandemic, so the data cover the period from early to later on in the pandemic process. Furthermore, the self-reported health survey method we have used is highly validated and commonly used in research in the quality of life field. 

## Figures and Tables

**Figure 1 ijerph-20-06397-f001:**
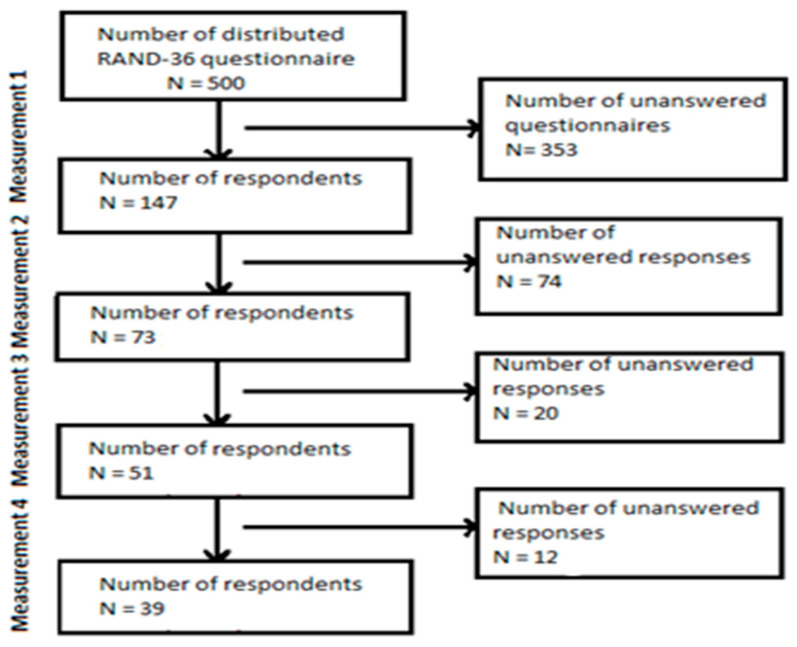
Response rate.

**Table 1 ijerph-20-06397-t001:** Demographic characteristics of participants at each measurement.

Measurement Characteristic	Month 0 (n = 147)	Month 4 (n = 73)	Month 8 (n = 51)	Month 12 (n = 39)
Gender:				
Female (n, %)	120 (81.5)	61 (83.6)	42 (82.4)	34 (87.2)
Male (n, %)	27 (18.5)	12 (16.4)	9 (17.6)	5 (12.8)
Years mean (n)	38.2	39.5	40.4	40.0
Working years mean (n)	13	13	15	15
Category group				
Physician (n, %)	22 (15.1)	10 (13.7)	5 (9.8)	5 (12.8)
Registered nurse (n, %)	64 (44.5)	39 (53.4)	29 (56.9)	23 (59.0)
Licensed practical nurses (n, %)	61 (40.4)	24 (32.9)	17 (33.3)	11 (28.2)
Type of Ward				
Infection ward (n, %)	31 (21.2)	14 (19.2)	6 (11.8)	4 (10.3)
The intensive care unit (n, %)	30 (20.5)	19 (26.0)	14 (27.5)	12 (30.8)
Orthopedic department (n, %)	21 (14.5)	13 (17.8)	11 (21.6)	5 (12.8)
Thoracic clinic (n, %)	23 (15.7)	12 (16.4)	10 (19.6)	9 (23.1)
The department of cardiology (n, %)	42 (28.2)	15 (20.5)	10 (19.6)	9 (23.1)

**Table 2 ijerph-20-06397-t002:** Comparison of perceived level of QoL for the whole group; measurements one to four, showing a significant change over time.

RAND-36									
	Month 0	Month 4	*p* Value	Month 4	Month 8	*p* Value	Month 8	Month 12	*p* Value
Physical functioning (PF)	93.25	91.49	0.391	91.49	89.70	0.307	89.70	92.56	0.072
Role functioning/physical (RP)	71.37	71.91	0.830	71.91	76.00	0.907	76.00	76.92	0.903
Bodily pain (BP)	79.63	77.28	0.585	77.28	82.03	0.552	82.03	77.18	0.050
General health (GH)	73.78	67.39	0.003	67.39	71.11	0.497	71.11	71.23	0.163
Energy/fatigue (VT)	51.75	48.21	0.006	48.21	47.54	0.883	47.54	47.69	0.148
Social functioning (SF)	66.63	67.80	0.963	67.80	70.78	0.802	70.78	71.02	0.604
Role functioning/emotional (RE)	64.59	63.93	0.166	63.93	58.80	0.112	58.80	61.53	0.877
Emotional well-being (MH)	71.73	67.90	0.014	67.90	71.33	0.524	71.33	70.67	0.542
Responders	147	73		73	51		51	39	

**Table 3 ijerph-20-06397-t003:** Pandemic ward vs. non-pandemic ward differences; mean for months 0 and 4.

RAND-36	Month 0		Month 4		*p* Value
	Pandemic	Non-Pandemic Wards	Pandemic	Non-Pandemic Wards	
Physical functioning (PF)	93.36	93.44	90.91	94.04	0.353
Role functioning/physical (RP)	65.57	75.00	66.67	83.33	0.073
Bodily pain (BP)	75.91	86.05	73.97	80.74	0.150
General health (GH)	74.23	75.00	64.39	73.33	0.100
Energy/fatigue (VT)	46.67	56.25	38.79	58.33	0.001
Social functioning (SF)	65.42	67.58	59.97	77.00	0.172
Role functioning/emotional (RE)	62.29	67.72	55.57	64.15	0.200
Emotional well-being (MH)	71.75	71.62	64.39	70.81	0.229
Responders	60	63	33	27	

**Table 4 ijerph-20-06397-t004:** Quality of life comparison by groups of profession, divided into professions with significant differences in QoL.

RAND-36	Mean	95% CI	*p* Value
Physical functioning (PF)	93.25	91.5–94.9	0.046
Role functioning/physical (RP)	71.38	65.1–77.6	0.683
Bodily pain (BP)	79.64	75.8–83.4	0.013
General health (GH)	73.78	70.7–76.9	0.737
Energy/fatigue (VT)	51.76	48.2–55.3	0.297
Social functioning (SF)	66.64	62.4–70.8	0.002
Role functioning/emotional (RE)	64.59	58.2–70.9	0.016
Emotional well-being (MH)	71.73	68.9–74.5	0.101
Profession	Physician mean	Registered nurses mean	Licensed practical nurses mean
Physical functioning (PF)	94.54	95.15	90.67
Bodily pain (BP)	88.63	82.38	73.30
Social functioning (SF)	81.81	67.97	59.53
Role functioning/emotional (RE)	86.36	60.51	60.91

## Data Availability

The data presented in this study are available on request from the corresponding author.

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
