# Peer review of "Quality of Life in Healthcare Workers during COVID-19—A Longitudinal Study"

_ijerph, 2023, doi:10.3390/ijerph20146397_

Round 1
Reviewer 1 Report
Dear author, I have reviewed the paper "Quality of life in healthcare workers during Covid-19 - a longitudinal study", which aimed to examine healthcare workers' quality of life during the Covid-19 pandemic. The author prior to further processing his paper should focus efforts on the following comments:
1. The first sentence of the abstract is very terse and does not put the problem in context, prior to describing the objective.
2. Authors should avoid repeating the words of the title in key words.
3. The introduction should contain at least 4 paragraphs where it addresses the current situation of the object of study at all levels; the problem and its objective in the last paragraph.
4. In the conclusions it is necessary to include a last paragraph on the limitations of the study and future studies based on its results.
5. The limitations written in discussion should be transferred to the conclusion.
6. The wording of the authors regarding the size of the paragraphs should be very extensive; they should necessarily generate paragraphs of between 100-180 words.
7. The methodology and processing is very clear.
In the current form and with minor corrections I consider it is possible to publish the manuscript. The authors should strengthen a more orderly introduction. I suggest to develop between 4 - 6 paragraphs. Finally, reorder the discussion considering the objectives and/or research question.
--
Author Response
Dear author, I have reviewed the paper "Quality of life in healthcare workers during Covid-19 - a longitudinal study", which aimed to examine healthcare workers' quality of life during the Covid-19 pandemic. The author prior to further processing his paper should focus efforts on the following comments:
Ridell: Thank you for reading our article and your valuable comments!
The first sentence of the abstract is very terse and does not put the problem in context, prior to describing the objective.
Ridell: Thank you for this point, we have re-written and tryed to make a better context in the abstract.
Authors should avoid repeating the words of the title in key words.
Ridell: A good point aswell, we have changed the keywords for this.
The introduction should contain at least 4 paragraphs where it addresses the current situation of the object of study at all levels; the problem and its objective in the last paragraph.
Ridell: Thank you. We have tryed to rearrange the introduction and refilled it.
In the conclusions it is necessary to include a last paragraph on the limitations of the study and future studies based on its results.
The limitations written in discussion should be transferred to the conclusion.
Ridell: Good comment, we moved and re-written the paragraph to the conclusion instead of discussion.
The wording of the authors regarding the size of the paragraphs should be very extensive; they should necessarily generate paragraphs of between 100-180 words.
The methodology and processing is very clear.
Ridell: Thank you!
In the current form and with minor corrections I consider it is possible to publish the manuscript. The authors should strengthen a more orderly introduction. I suggest to develop between 4 - 6 paragraphs. Finally, reorder the discussion considering the objectives and/or research question.
Ridell: Thank you for valuable comments. We hope our revision of the manuscript now meets the requirements for a publication.
Reviewer 2 Report
What RAND-36 is should be explained better in the introduction.
The number of respondents responding to the survey is usually one-third in each period. What kind of method should be followed to increase the number of participants? The number of participants is too low to comment.
The discussion section should be written more methodologically. I suggest that comparative data be grouped and divided into more paragraphs.
Author Response
What RAND-36 is should be explained better in the introduction.
Ridell: Thank you for your comment! We tryed to developed the description.
The number of respondents responding to the survey is usually one-third in each period. What kind of method should be followed to increase the number of participants?
Ridell: Good question, during the pandemic the healthcareworkers was deeply exhausted and the fatigue that seems to have existed may have reduced the priority of filling in the survey.
The number of participants is too low to comment.
Ridell: This is a good point and comparison between groups was not the main aim and we agree with the reviewer. We tryed to re-written for avoiding conclusions regarding the last measurement and the small amount of participants.
The discussion section should be written more methodologically. I suggest that comparative data be grouped and divided into more paragraphs.
Ridell: Thank you for this point of view, we tryed to rearranged so it´s easier to follow the paragraphs.
Reviewer 3 Report
In the manuscript “Quality of Life in Healthcare Workers during Covid-19 – A Longitudinal Study”, the authors present and discuss the impact of Covid19 pandemic over different types of healthcare workers in different practice settings in southeast Sweden during 12 months, accessed through questionnaires every three months.
Data correlates with similar studies, though it was only partially presented. It would be interesting to also present the stratified data for the third and fourth measurements, validating your statements about them.
Avoid presenting data like table 5, that as you had mentioned are probably not comparable as the data from mean general population was accessed in a different context, when there was no pandemics and all its associated impacts in everyone’s life.
Avoid using non-English acronyms such as VT for energy/fatigue or MH for emotional well-being.
Minor typos
Author Response
In the manuscript “Quality of Life in Healthcare Workers during Covid-19 – A Longitudinal Study”, the authors present and discuss the impact of Covid19 pandemic over different types of healthcare workers in different practice settings in southeast Sweden during 12 months, accessed through questionnaires every three months.
Ridell: Thank you kindly for your comments.
Data correlates with similar studies, though it was only partially presented. It would be interesting to also present the stratified data for the third and fourth measurements, validating your statements about them.
Ridell: A very good point! We have made a new table with all four measurements and p-value inbetween for validating the statements.
Avoid presenting data like table 5, that as you had mentioned are probably not comparable as the data from mean general population was accessed in a different context, when there was no pandemics and all its associated impacts in everyone’s life.
Ridell: We agree with the rewiever, but it would possibly be interesting to show a value on the population´s well-being before the pandemic, because healthcare workers have better values even during the first measurement occasion of covid-19 in physical characteristics. We removed table 5 but discuss the study instead.
Avoid using non-English acronyms such as VT for energy/fatigue or MH for emotional well-being.
Ridell: The acronyms as VT and MH is standard for the RAND-36 instrument, which we could be more clear about. Thank you for pointing this out!
Reviewer 4 Report
Dear authors, I have read your paper with great interest. The topic was very interesting and informative. But I had several concerns about this manuscript. Unfortunately, the methodological work is unacceptable to me. The study design is not clearly described. It is not clear which hospital is in question and how many employees it has. It has not been stated how the anonymity of the respondents was protected since you interviewed them multiple times. The sample is based on 39 respondents. In addition, there is a large difference considering gender and employee structure, thus all conclusions are irrelevant. It is unclear why so few people agreed to participate in the study and why so many were dropping out over time. I feel that the present article could not be published in its present form.
Author Response
Dear authors, I have read your paper with great interest. The topic was very interesting and informative.
Ridell: Thank you!
But I had several concerns about this manuscript.
Unfortunately, the methodological work is unacceptable to me.
Ridell: We have now tried to make the methodological more clarified.
The study design is not clearly described.
Ridell: Thank you for this point. We have re-written the design and hope it is clearer now.
It is not clear which hospital is in question and how many employees it has.
Ridell: We are not used to precisely point out what hospital that is in the question. This is especially important when the included wards and study cohort can be identified. This is therefore we just inform that it is “at a larger university hospital in the southeast of Sweden”. However, we have added the information about how many employees it has.
It has not been stated how the anonymity of the respondents was protected since you interviewed them multiple times.
Ridell: Good comment. The respondents answered anonymously on the paper form of the instrument RAND-36 and was not interviewed. The survey was coded with the aim that only those who answered on all occasions would be included in the final analysis. We have tried to clarify the text to accomplish this.
The sample is based on 39 respondents. In addition, there is a large difference considering gender and employee structure, thus all conclusions are irrelevant.
Ridell: This is a very good point. Our attempt with the study is to examine healthcare workers’ quality of life with RAND-36 during one year of the Covid-19 pandemic, not to compare the results between the different groups. We agree with the reviewer and have tried to alter the text mainly in the discussion section in the manuscript to accomplish this.
It is unclear why so few people agreed to participate in the study and why so many were dropping out over time.
Ridell: Also, a good point. As the respondents participated voluntarily and anonymously in the study, we couldn’t ask them the reason for dropouts. That is why we tried to have an explanation of this in the discussion section (see page 09, line 273).
I feel that the present article could not be published in its present form.
Ridell: Thank you for valuable comments. We hope our revision of the manuscript now meets the requirements for a publication.
Round 2
Reviewer 4 Report
The manuscript has been significantly improved. All suggestions have been accepted.